

# Identification of a five-miRNA signature predicting survival in cutaneous melanoma cancer patients

Tao Lu[1,2,3], Shuang Chen[1], Le Qu[1], Yunlin Wang[1], Hong-duo Chen[1] and Chundi He[1,3]

[1] Department of Dermatology, No. 1 Hospital of China Medical University, Shenyang, Liao Ning, China
[2] Department of Dermatology, Affiliated Hospital of Chifeng University, Chifeng, Inner Mongolia, China
[3] Graduate school, China Medical University, Shenyang, Liao Ning, China

## ABSTRACT

**Background**. Cutaneous melanoma (CM) is the deadliest form of skin cancer. Numerous studies have revealed that microRNAs (miRNAs) are expressed abnormally in melanoma tissues. Our work aimed to assess multiple miRNAs using bioinformatic analysis in order to predict the prognoses of cutaneous melanoma patients.

**Methods**. The microarray dataset GSE35579 was downloaded from the Gene Expression Omnibus (GEO) database to detect the differential expression of miRNAs (DEMs), including 41 melanoma (primary and metastatic) tissues and 11 benign nevi. Clinical information and miRNA sequencing data of cutaneous melanoma tissues were downloaded from the Cancer Genome Atlas database (TCGA) to assess the prognostic values of DEMs. Additionally, the target genes of DEMs were anticipated using miRanda, miRmap, TargetScan, and PicTar. Finally, functional analysis was performed using selected target genes on the Annotation, Visualization and Integrated Discovery (DAVID) website.

**Results**. After performing bioinformatic analysis, a total of 185 DEMs were identified: 80 upregulated miRNAs and 105 downregulated miRNAs. A five-miRNA (miR-25, miR-204, miR-211, miR-510, miR-513c) signature was discovered to be a potential significant prognostic biomarker of cutaneous melanoma when using the Kaplan–Meier survival method ($P = 0.001$). Univariate and multivariate Cox regression analyses showed that the five-miRNA signature could be an independent prognostic marker ($HR = 0.605$, $P = 0.006$) in cutaneous melanoma patients. Biological pathway analysis indicated that the target genes may be involved in PI3K-Akt pathways, ubiquitin-mediated proteolysis, and focal adhesion.

**Conclusion**. The identified five-miRNA signature may serve as a prognostic biomarker, or as a potential therapeutic target, in cutaneous melanoma patients.

Corresponding author
Chundi He, cdhe@cmu.edu.cn

## INTRODUCTION

Melanoma is the result of the malignant transformation of melanocytes. It accounts for approximately only 5% of all skin malignancies, but is thought to be the most invasive and lethal form (*Dimitriou et al., 2018*). When compared with non-melanoma skin cancer,

melanoma has higher invasiveness and a worse prognosis, and its incidence has increased significantly in recent years (*Ross et al., 2018*). The histopathological features of melanoma are not as distinguishing as its molecular heterogeneity, and its formation is based on the continuous alteration of specific genes and pathways that control metabolism or regulate key cellular functions (*Palmieri et al., 2018*). Although new treatments for melanoma are constantly being developed, their effectiveness is still unsatisfactory. If metastasis occurs, the results could be life-threatening, prompting clinicians to pursue new predictive markers and targeted therapeutic genes (*Falzone, Salomone & Libra, 2018*; *Leonardi et al., 2018*; *Perera et al., 2013*).

MicroRNAs (miRNAs) are small single-stranded non-coding RNAs with oncogenic or tumor-suppressive roles, consisting of 20–26 nucleotides. There is growing evidence that miRNAs play key roles in multiple developmental stages of human cancers, including cutaneous melanoma (*Gulyaeva & Kushlinskiy, 2016*). These molecules can affect gene transcription by complementing the promoter region of a particular gene or by directly regulating the activity of a gene (*Piletic & Kunej, 2016*). miRNAs can even be released into the bloodstream by a tumor and can be detected in melanoma cells. Dysregulated miRNAs can also be used as survival markers in cutaneous melanoma patients (*Riefolo et al., 2019*), as seen in the signature of 18 miRNAs identified by *Segura et al. (2010)* and the miRNAs identified by *Caramuta et al. (2010)*.

With the development of sequencing technology and various omics (genomics, transcriptomics, proteomics, etc.), a large amount of biological data have been acquired, leading to the development of bioinformatics to mine, utilize and integrate these data (*Manzoni et al., 2018*). Bioinformatics can be defined as the practical discipline of applying informatics technology (including applied mathematics, computer science and statistics) to understand and process biological tissue information related to macromolecules on a large scale (*Luscombe, Greenbaum & Gerstein, 2001*). It has been applied in various fields of biological research, such as in the identification or diagnosis of disease biomarkers (*Shergalis et al., 2018*), individualized treatment of cancer, the preparation of tumor vaccines (*Hu, Ott & Wu, 2018*), among others. Various large public bioinformatics databases have been developed, such as the Gene Expression Omnibus (GEO, https://www.ncbi.nlm.nih.gov/geo/) database, which is a functional genomics data repository that help users download experiments and curate gene expression profiles (*Barrett et al., 2013*), and the Cancer Genome Atlas (TCGA, https://portal.gdc.cancer.gov/), which is the most useful tumor genomics program with at least 30 cancer types included (*Chandran et al., 2016*). These two databases play a wide range of roles in the research of cancers including melanoma. For example, the data provided by GEO are used in the screening of melanoma prognostic factors (*Hu et al., 2019*; *Xin et al., 2019*). *Robertson et al. (2017)* conducted an in-depth analysis of UM (uveal melanoma) samples from TCGA project in order to gain a deeper understanding of the biological processes of UM tumors with distinct prognoses. Even anti-tumor drugs in melanoma patients with or without anti-PD-1 treatment have been analyzed using the GEO and TCGA databases (*Wu et al., 2019*).

Previous studies have shown that multiple miRNAs can be used as diagnostic and prognostic markers, but the number of cases is relatively small with inconsistent results (*Jayawardana et al., 2016*; *Ross et al., 2018*). In our study, we screened microarray data from GEO, then downloaded the clinical information and expression profiles of cutaneous melanoma patients from TCGA. Our aim was to use bioinformatics methods on a large sample from TCGA database, evaluate the prognostic value of the differential expression of miRNAs (DEMs) by analyzing the high-throughput sequencing data, and establish a five-miRNA predictive signature of patient survival.

## MATERIALS & METHODS

### Acquisition of a microarray data source and DEMs

Variations in sample collection, storage, batch testing and detection platform methods can lead to data deviation, and so we could not simply directly combine different datasets. When selecting datasets, the inclusion criteria were: (1) datasets including more than 40 samples; (2) datasets with tumor tissue samples; and (3) datasets containing melanoma (including primary and metastatic melanoma) and pigmented nevi tissue simultaneously. Ultimately, GSE35579 (cutaneous melanoma = 41, nevus = 11) was the one that met our requirements. After microarray data were downloaded from the GEO of the National Center for Biotechnology Information (NCBI), we preprocessed the miRNA expression data using the quantile normalized method, then identified DEMs between melanoma and benign nevi. The "limma" package in software R (*R Core Team, 2018*), which can be operated flexibly offline, was selected to process the differential expression data. $P < 0.05$ was selected as the statistically significant cut-off criterion.

### Identification of the relationship between DEMs and overall survival (OS) in melanoma patients

DEMs obtained from the GEO database were matched in TCGA's database for further analysis of their relationship to patient prognosis. After removing patients without completed clinical information and with an OS (overall survival) time <1 month, the remaining 428 patients were divided into high-risk and low-risk groups according to the median score of DEM expression level. We used 120 months as the end of our observation time, and patients with a survival period of more than 120 months were considered survivors. The DEM expression profiles acquired from TCGA, after being log2 transformed, were assessed with the Kaplan–Meier method and a log-rank test to preliminarily identify the miRNAs that were associated with patient survival. $P < 0.05$ was considered to indicate statistically significant differences. Considering that multiple miRNAs have more reliable predictive effects, we used different combinations of several prognosis-related miRNAs to calculate the risk score for each melanoma patient in accordance with a high or low level of expression. Using this method, we classified the patients into two groups, high and low score groups, further using Kaplan–Meier analysis to assess these miRNA combinations. To make the experiment more objective, we only selected a meaningful combination of DEMs for follow-up study. The Kaplan–Meier results were visualized by the GraphPad

Prism 5.0. Finally, univariate and multivariate Cox regression analyses were conducted to verify the prognostic role of the screened DEM signature.

### Prediction of the DEM signature's target genes

TargetScan (http://www.targetscan.org/), miRDB (http://www.mirdb.org/), DIANA (http://www.microrna.gr/microT-CDS), and miRmap (https://mirmap.ezlab.org/app/) were used to predict these candidate target genes. The FunRich (Functional enrichment analysis (http://www.funrich.org/) tool was used to intersect the results of the four prediction tools for further research.

### Analysis of functions and pathways of target genes

GO term and KEGG pathway analysis was conducted on the overlapping terms in the Database for Annotation, Visualization and Integrated Discovery (DAVID, https://david.ncifcrf.gov/home.jsp) websites, providing a comprehensive set of functional annotation tools to help determine the biological meaning of the genes. The results were visualized by the R "ggplot2" package. A $P$-value $< 0.05$ was set as the cut-off criterion.

### Statistical analysis

To compare the miRNA expression data between melanomas and nevi, we used unpaired $t$-tests. The relationship between expression data and the melanoma patients' clinical information was assessed with chi-square and $t$-tests. Kaplan–Meier survival analysis and univariate/multivariate Cox proportional hazard regression analyses were performed using IBM SPSS Statistics 25.0 to assess each DEM and miRNA signature prognostic function. A $P$-value $< 0.05$ was considered statistically significant.

## RESULTS

### DEM identification and construction of a five-miRNA signature

We first acquired 185 (80 upregulated and 105 downregulated) differentially expressed miRNAs between melanomas (primary and metastatic) and benign nevi from the miRNA dataset GSE35579, according to the standard of $P < 0.05$. To make the results more intuitive, we used volcano maps (Fig. 1) to show these DEMs. To find out whether differential miRNAs could distinguish a cancer sample from a nevus sample, we selected the top 100 miRNAs with the most obvious differences in expression to perform a hierarchic cluster heat map based on Euclidean distance (as shown in Fig. 2), and the results were acceptable.

DEMs were matched in TCGA, and combined with the clinical information of melanoma patients (shown in Table 1). DEMs that could not be found in TCGA were excluded, and each miRNA was analyzed to determine whether it was related to the prognoses. According to our statistical analysis, there were eight genes (miR-25, miR-100, miR-204, miR-211, miR-19b, miR-510, miR-511, miR-513c) closely related to the prognoses of the patients.

We integrated the expression of these eight miRNAs and applied different combinations calculating the risk score for each patient (the details of which can be found in the Methods section), and we found a very obvious prognostic significance when five of the miRNAs
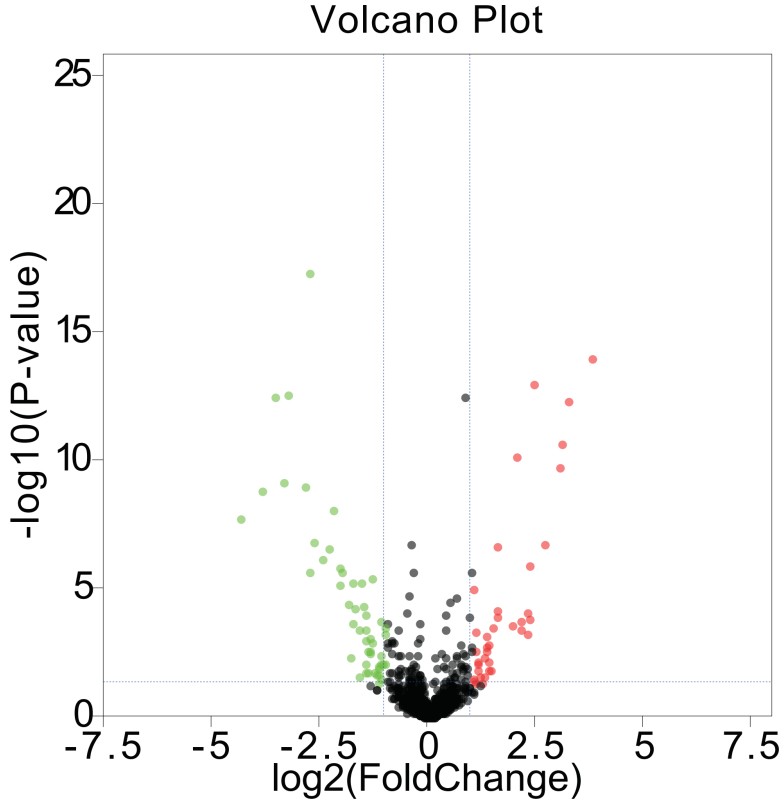

## Volcano Plot

**Figure 1  Volcano maps.** DEMs in GSE35579 between melanoma and nevi: green represents down regulation in melanoma (logFC < −1); red represents up regulation in melanoma (logFC > 1); $P < 0.05$.

(miR-25, miR-204, miR-211, miR-510, miR-513c) were combined. Among these miRNAs, miR-204 correlated positively with prognosis, while miR-25, miR-211, miR-510, and miR-513c correlated negatively with prognosis. These details are listed in Table 2. We then established a five-miRNA-based prognostic model. The prognostic characteristics of these five miRNAs are shown in Fig. 2 (Figs. 3A–3E). According to the median risk score, 428 patients were divided into a high-risk group ($n = 222$) and a low-risk group ($n = 206$). Survival analysis was performed using the Kaplan–Meier method with the log-rank test. The results showed that the survival rate of the high-risk group was significantly lower than that of the low-risk group ($P = 0.001$, Fig. 3F). To explore whether these five miRNAs were related to clinical features, we conducted correlation analysis and found that, except for miR-25, these miRNAs might be related to T stage and Breslow depth value (see Table 3 for details).

Univariate and multivariate Cox regression analyses were used to test the effects of the five miRNA features (high-risk and low-risk) combined with the clinical features of melanoma patients (including age, T stage, N stage, M stage, and clinical stage) on OS (overall survival). Univariate analysis showed that T stage (HR = 0.443, $P < 0.000$), N stage (HR = 0.517, $P < 0.000$), clinical stage (HR = 0.543, $P < 0.000$), age at diagnosis (HR = 1.563, $P = 0.003$), Breslow depth value (HR = 0.370, $P < 0.000$) and the five-miRNA
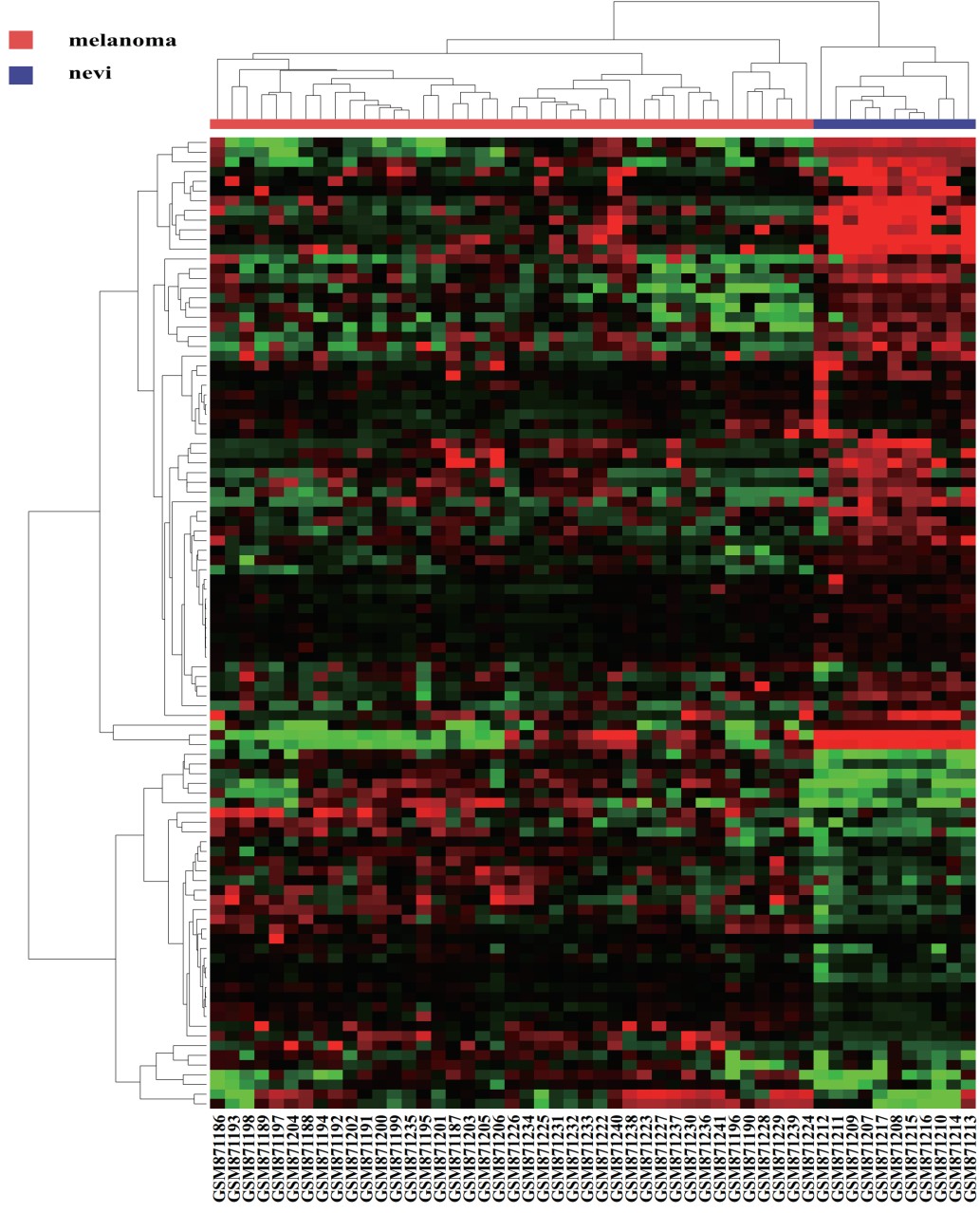

**Figure 2  A hierarchic figure.** Hierarchic analysis: green, black and red indicate the top 100 downregulated, nonsignificantly differentially expressed and up regulated DEMs, Respectively.

characteristics (HR = 0.613, $P = 0.001$) were associated with OS in melanoma patients. In the multivariate analysis, the five-miRNA characteristics (HR = 0.605, $P = 0.006$), N stage (HR = 0.246, $P = 0.008$) and Breslow depth value (HR = 0.509, $P = 0.011$) were all shown to be independent prognostic factors for patients with cutaneous melanoma (Table 4).

**Table 1   Clinical feature of melanoma patients from TCGA.**

| Variables | Case, $n$ (%) |
|---|---|
| Age at diagnosis (yr) | |
| <60 | 241 |
| ≥60 | 221 |
| NA | 8 |
| Gender | |
| Male | 290 |
| Female | 180 |
| T stage | |
| T0 | 23 |
| T1 ($a+b$) | 42 |
| T2 ($a+b$) | 78 |
| T3 ($a+b$) | 90 |
| T4 ($a+b$) | 143 |
| Tis | 8 |
| TX | 47 |
| NA | 29 |
| Pathologic stage | |
| Stage 0 | 7 |
| Stage I | 77 |
| Stage II | 140 |
| I/II NOS | 14 |
| Stage III | 171 |
| Stage IV | 23 |
| NA | 38 |
| Node status | |
| N0 | 235 |
| N1–3 | 178 |
| Nx | 36 |
| NA | 21 |
| Metastasis | |
| M0 | 418 |
| M1 | 24 |
| NA | 28 |

**Notes.**
    NA,  Not available.

## GO and KEGG pathway analysis of target genes of the five-miRNA signature

Using TargetScan, miRDB, DIANA, and miRmap, we predicted five miRNA target genes (miR-25, miR-204, miR-211, miR-510, and miR-513c). The four prediction tools' intersecting genes were visualized using the FunRich tool (Figs. 4A–4E). A total of 367 (228 miR-25, 28 miR-204, 31 miR-211,77 miR-510, 37 miR-513c, and deleted duplicate values) overlapping genes were entered into DAVID, and assessed by GO and KEGG pathway analyses. The biological process (BP) analysis results showed that the genes were

**Table 2** The prognostic related differentially expressed miRNAs identifed between melanoma and nevi.

| Down-regulation DEMs | *P*-value | Up-regulation | *P*-value |
| --- | --- | --- | --- |
| hsa-miR-211 | 0.033 | hsa-miR-25 | 0.004 |
| hsa-miR-204 | 0.001 | | |
| hsa-miR-510 | 0.007 | | |
| hsa-miR-513c | 0.014 | | |

**Notes.**
DEMs, Differentially expressed miRNAs.

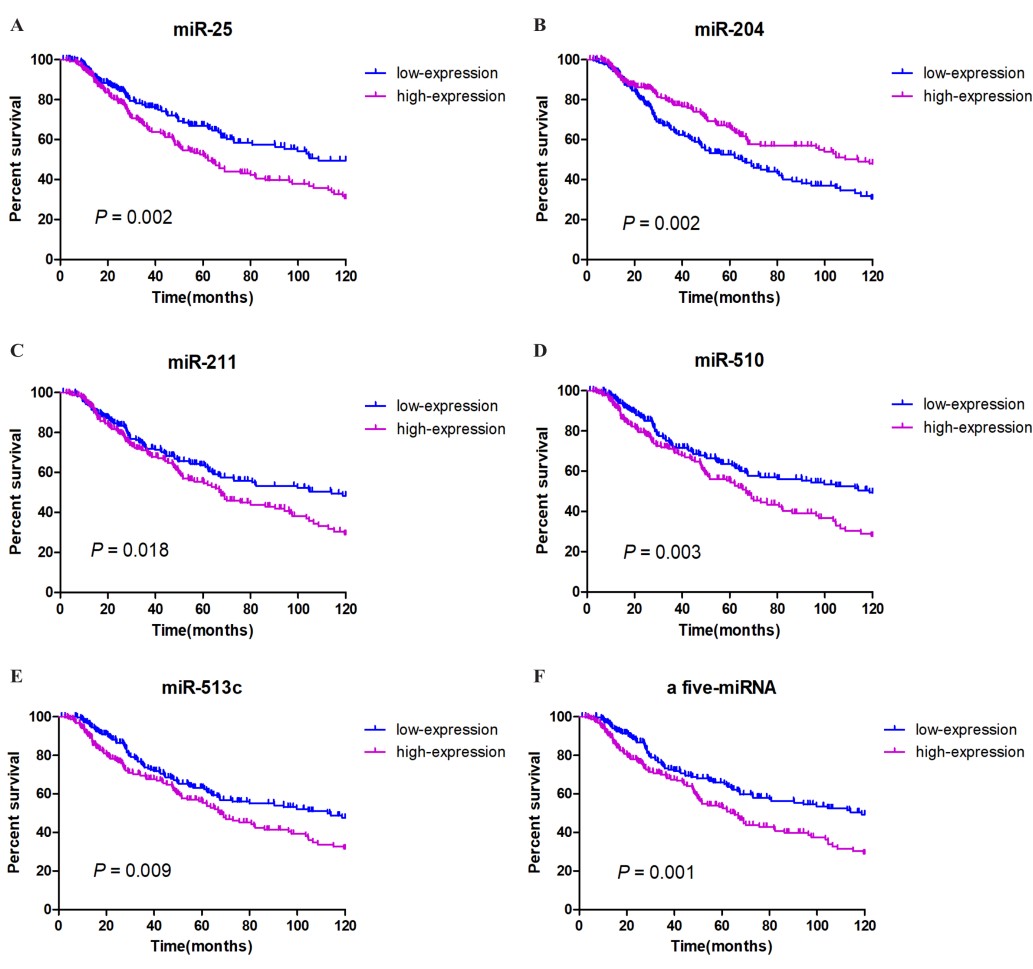

**Figure 3** Survival analysis and prognosis of cutaneous melanoma patients from TCGA. A–E show that the prognostic characteristics of these five miRNAs (miR-25, miR-204, miR-211, miR-510, miR-513c); F shows that the prognostic characteristics of a 5-miRNA signature (miR-25, miR-204, miR-211, miR-510, miR-513c). Blue curves represent the low expression group, and purple curves represent the high expression group.

concentrated in the transcription regulation, angiogenesis, protein phosphorylation and ubiquitination (Fig. 5A). Cellular component (CC) analysis showed that these genes were concentrated in the cytoplasm, nucleus, focal adhesion and cell–cell adherens junctions

Lu et al. (2019), *PeerJ*, DOI 10.7717/peerj.7831

**Table 3** Association between the five miRNAs and melanoma cancer clinical characters.

| Variables | miR-25 expression | | | miR-204 expression | | | miR-211 expression | | | miR-510 expression | | | miR-513c expression | | |
|---|---|---|---|---|---|---|---|---|---|---|---|---|---|---|---|
| | Low | High | P | Low | High | P | Low | High | P | Low | High | P | Low | High | P |
| Age | | | | | | | | | | | | | | | |
| <60 | 124 | 106 | 0.064 | 92 | 138 | <0.000* | 122 | 108 | 0.175 | 119 | 111 | 0.379 | 115 | 115 | 0.755 |
| ≥60 | 89 | 109 | | 118 | 80 | | 92 | 106 | | 94 | 104 | | 96 | 102 | |
| Gender | | | | | | | | | | | | | | | |
| Female | 80 | 81 | 0.980 | 83 | 78 | 0.424 | 73 | 88 | 0.134 | 76 | 85 | 0.411 | 73 | 88 | 0.203 |
| Male | 133 | 134 | | 127 | 140 | | 141 | 126 | | 137 | 130 | | 138 | 129 | |
| T stage | | | | | | | | | | | | | | | |
| T1 + T2 | 75 | 62 | 0.157 | 54 | 83 | 0.004* | 84 | 53 | <0.000* | 79 | 58 | 0.003* | 75 | 62 | 0.037* |
| T3 + T4 | 103 | 116 | | 121 | 98 | | 89 | 130 | | 91 | 128 | | 95 | 124 | |
| Lymph node status | | | | | | | | | | | | | | | |
| N0 | 106 | 108 | 0.758 | 107 | 107 | 0.955 | 107 | 107 | 0.955 | 109 | 105 | 0.811 | 103 | 111 | 0.516 |
| N1–2 | 82 | 89 | | 86 | 85 | | 85 | 86 | | 85 | 86 | | 88 | 83 | |
| Mestasis | | | | | | | | | | | | | | | |
| M0 | 192 | 193 | 0.841 | 189 | 196 | 0.769 | 188 | 197 | 0.243 | 193 | 192 | 0.823 | 188 | 197 | 0.458 |
| M1 | 10 | 11 | | 11 | 10 | | 13 | 8 | | 10 | 11 | | 12 | 9 | |
| Clinical stage | | | | | | | | | | | | | | | |
| I + II | 107 | 95 | 0.166 | 102 | 100 | 0.715 | 97 | 105 | 0.512 | 97 | 105 | 0.738 | 93 | 109 | 0.250 |
| III + IV | 84 | 99 | | 89 | 94 | | 94 | 89 | | 91 | 92 | | 95 | 88 | |
| Breslow depth value (mm) | | | | | | | | | | | | | | | |
| <3 | 83 | 77 | 0.255 | 70 | 90 | 0.041* | 91 | 69 | <0.000* | 91 | 69 | <0.000* | 84 | 76 | 0.015* |
| ≥3 | 78 | 93 | | 94 | 77 | | 61 | 110 | | 61 | 110 | | 67 | 104 | |

**Notes.**

*$P < 0.05$ was considered statistically significant.

**Table 4  TCGA univariable and multivariable Cox regression analysis.**

| Variables | Univariate analysis | | Multivariate analysis | |
|---|---|---|---|---|
| | HR (95% CI) | P value | HR (95% CI) | P value |
| Age at diagnosis (≥60 vs. <60) | 1.563 (1.166–2.096) | 0.003[*] | 1.199(0.839–1.714) | 0.319 |
| T stage (T3 + T4 vs. T1 + T2) | 0.443(0 .316-0.622) | <0.000[*] | 0.825(0.468–1.456) | 0.507 |
| N stage (N1–2 vs. N0) | 0.517(0.380–0.703) | <0.000[*] | 0.246(0.087–0.699) | 0.008[*] |
| M stage (M1 vs. M0) | 0.554(0.292–1.051) | 0.071 | 0.616(0.218–1.741) | 0.361 |
| Clinical stage (III + IV vs. I + II) | 0.543(0.399–0.738) | <0.000[*] | 1.986(0.691–5.708) | 0.203 |
| Gender (Male vs. Female) | 0.997(0.734–1.355) | 0.986 | 1.040 (0.726–1.491) | 0.830 |
| Breslow depth value (≥3 mm vs. <3 mm) | 0.370(0.265–0.518) | <0.000[*] | 0.509(0.302–0.859) | 0.011[*] |
| Five-miRNA signature (high risk vs. low risk) | 0.613(0.456–0.823) | 0.001[*] | 0.605(0.424–0.863) | 0.006[*] |

**Notes.**

[*]$P < 0.05$ was considered statistically significant.

(Fig. 5B). Molecular function (MF) analysis showed that genes were concentrated in protein binding and ubiquitin-protein transferase activity (Fig. 5C). KEGG pathway analysis showed that these genes were concentrated in the PI3K-Akt signaling pathway, ubiquitin-mediated proteolysis, and focal adhesion pathway (Fig. 5D).

# DISCUSSION

The melanocyte pedigree is derived from the neural ridge, originating in the neural tube, and these cells migrate to specific locations in the skin , hair follicle and other parts of the body during embryonic development. In patients with vitiligo, the pigment ''island'' appears first in a location around the hair follicles after UVB treatment, which suggests that melanocytes have stem cell properties. These characteristics may contribute to the melanoma-derived invasion (*Mort, Jackson & Patton, 2015*). Caucasian populations have had a stable CM mortality rate since the 1990s, while the age-standardized mortality rate of cutaneous melanoma patients in East Asian populations has significantly increased over the past six decades (*Chen & Jin, 2016*). As the aging population grows, the incidence of melanoma is predicted to also increase (*Karimkhani et al., 2017*). Early detection and initial care are still crucial for treatment (*Schadendorf et al., 2018*). Recent studies have shown that epigenetic mechanisms play a very complex role in the development and progression of melanoma, including its methylation, chromosomal changes and remodeling, and regulation of the active function of various non-coding RNAs (*Sarkar et al., 2015*). Research on microRNAs is more developed compared to that of other non-coding RNAs, and these molecules play a significant role in nearly every biological process in nevi and melanomas, including proliferation, invasion, and apoptosis. Because of their chemical stability, these molecules can resist the degradation of RNase and can distinguish between different types of cancers, specifically those that can be secreted into the serum by tumor cells. Therefore, these molecules can be used to predict the prognosis and to perform the initial diagnosis in melanoma patients (*Ross et al., 2018*). Previous studies have explored the relationship between miRNAs and the prognoses of melanoma patients, but these studies were usually small sample studies without uniform results, that only looked at the primary melanoma

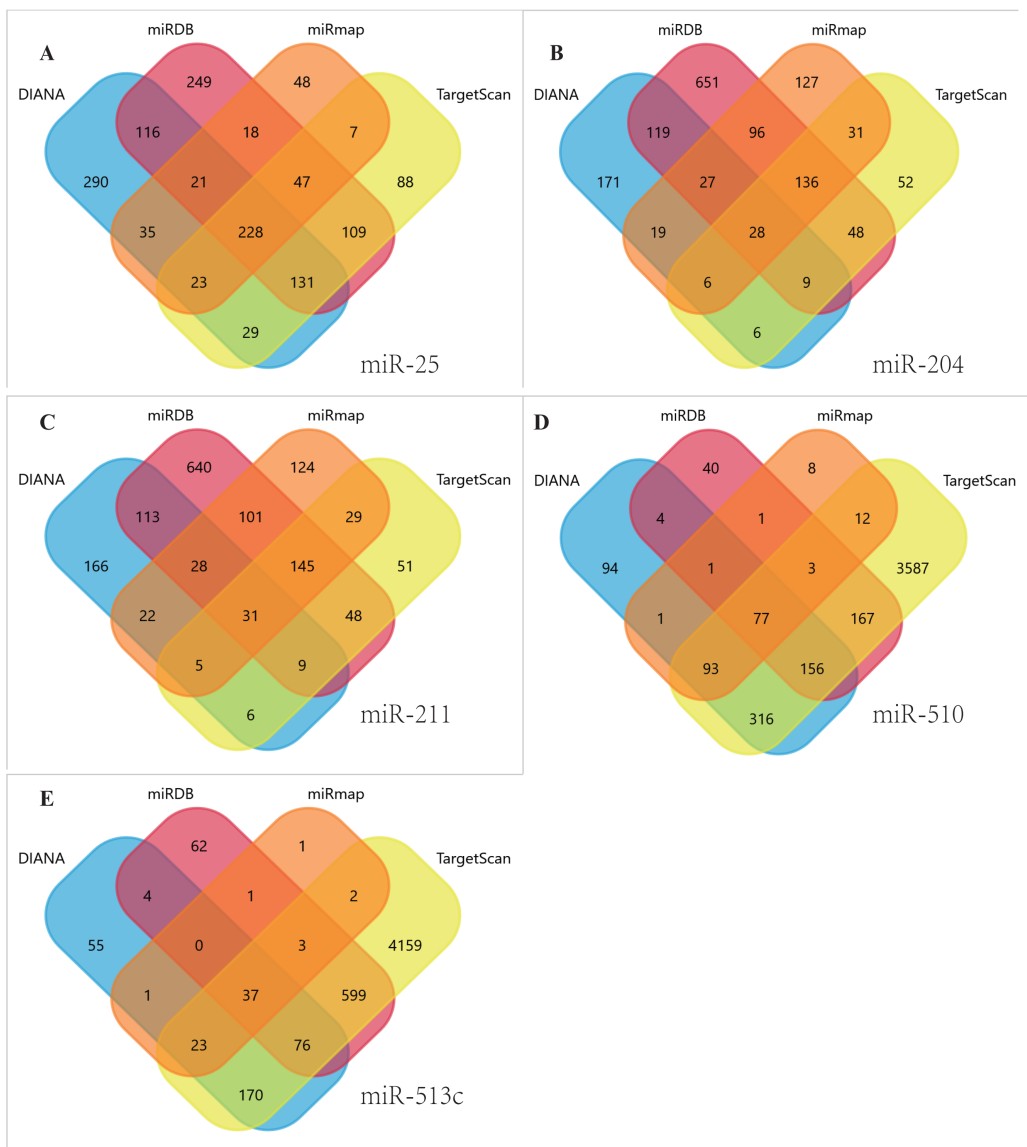

**Figure 4  Venn diagrams.** Target genes of the five DEMs were identified by four software programs, including TargetScan, miRDB, DIANA and miRmap, to acquire overlapping genes by the Funrich tool. (A) miR-25, (B) MiR-204, (C) MiR-211, (D) miR-510, (E) miR-513c.

or metastatic stage of the tumor, and only focused on single miRNAs (*Galasso et al., 2018*; *Jayawardana et al., 2016*; *Sanchez-Sendra et al., 2018*). TCGA's database contains an abundance of cancer information, making access to cancer expression profile data easy and cost-effective. However, unlike for other cancers, TCGA's database only contains skin melanoma samples without normal control tissue nor nevi tissue information, and we were unable to obtain differentially expressed miRNAs. Thus, we obtained the DEMs from the GEO database, performed prognostic analysis on the DEMs with expression profiles and clinical information sourced from TCGA's database, and gained eight prognosis-associated

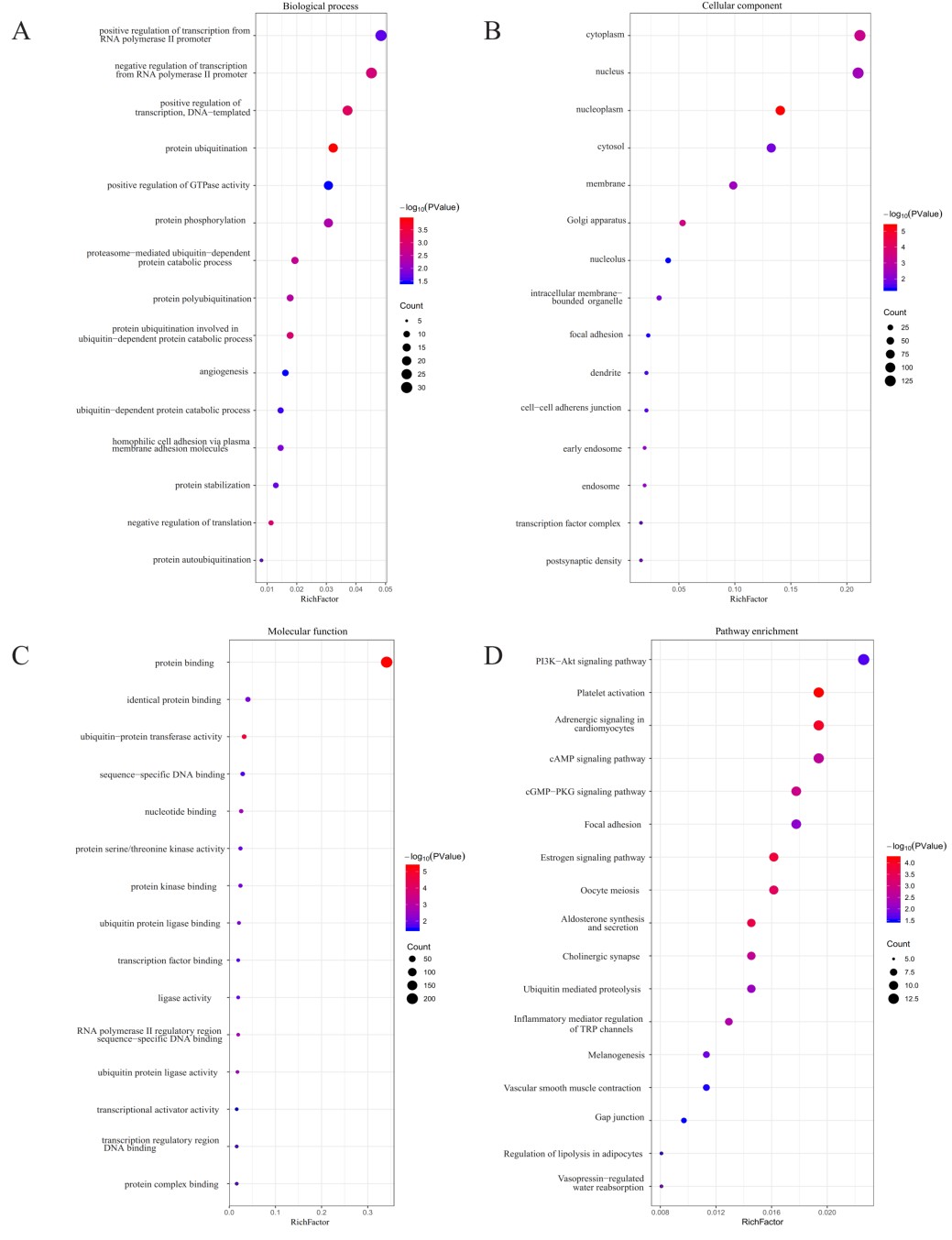

**Figure 5  GO and KEGG pathway analyses.** (A) The biological process (BP) analysis; (B) cellular component (CC) analysis; (C) molecular function (MF) analysis; (D) KEGG pathway analysis.

miRNAs. Considering that the main purpose of our experiment was to create a multi-gene-based reliable prognostic signature, and the different combinations of these miRNAs were not all meaningful, we will only discuss the five-miRNA signature that can be used as a prognostic factor to make our results more clear and purposeful. Through additional

analysis, we established a five-miRNA (miR-25, miR-204, miR-211, miR-510, miR-513c) signature to make the results more reliable. This five-miRNA based signature associated with the prognosis of cutaneous malignant melanoma will provide a theoretical basis for later studies, and can be used to non-invasively predict clinical prognoses in melanoma patients.

We used previous miRNA studies to verify the reliability of our results. In our study, miRNA-25 was the sole upregulated microRNA of the five miRNAs. *Zhu et al. (2014)* found that miRNA-25 can be secreted into serum and can be used as a marker for the early diagnosis of gastric cancer. *Kim et al. (2009)* found that miRNA-25 directly regulates P57 (a tumor suppressor gene), and the abnormal expression of this miRNA in gastric cancer patients can advance cancer cells from G1 to S phase. *Liu et al. (2012)* tested the serum of patients with pancreatic cancer and found that miRNA-25 levels were considerably higher than that of the normal control group, indicating that miRNA-25 can be used in the prognosis of pancreatic cancer. Zoni et al. observed that miR-25 is a key regulator of human prostate cancer invasiveness, and that it interacts directly with $\alpha(v)$—and $\alpha(6)$—integrins. Interestingly, *Zoni et al. (2015)* also found that miR-25 was a tumor suppressor in highly aggressive prostate cancer. *Huo et al. (2016)* found that the expression of miR-25 increased in melanoma tissue and melanocyte cell lines, and promoted melanoma cell proliferation and invasion, in part, by targeting Dickkopf-associated protein 3 (DKK3). This result is consistent with the poorer prognosis in melanoma patients with increased miRNA-25 expression. In our experiments, miRNA-204 was expressed at low levels in melanoma cells, and K-M survival analysis showed that the higher the expression, the longer the OS time. This finding is consistent with previous studies, such as Xin Chen et al.'s mechanistic study on the LINC01234-miR-204-5p-CBFB axis in gastric cancer patients where miR-204 was found to have low expression in gastric cancer tissues. In a gastric cancer cell miR-204 over exposure assay, miR-204 was found to inhibit the proliferation of cancer cells and to induce apoptosis and cell cycle arrest in G1-G0 phase, and survival analysis showed that patients with higher miR-204 levels had a better prognosis (*Chen et al., 2018*). In studies of colorectal cancer, nasopharyngeal carcinoma, and melanoma, miRNA-204 was suggested to have an inhibitory effect on cancer cells and to possibly be associated with cancer cell resistance or anti-radiation mechanisms (*Bian et al., 2016*; *Diaz-Martinez et al., 2018*; *Lu et al., 2016*). Marco et al. suggested that the low expression or loss of miR-204 play a key role in the progression of melanoma. In this study, miR-204 was associated with a better prognosis in cutaneous melanoma patients (*Galasso et al., 2018*). Previously published studies and data from existing miRNA databases indicate that miR-204-5p and miR-211-5p share some common targets, and, as a matter of fact, miR-204 and miR-211 have very similar nuclear targets. The nucleotide sequence has only two different nucleotides in the entire sequence and in the same seed region, which may be why these molecules share some common targets. In preceding studies, these miRNAs acted as tumor suppressors in melanoma and inhibited cell invasion (*Diaz-Martinez et al., 2018*; *Levy et al., 2010*). The most important function of miR-211 is the direct or indirect targeting of many other genes that may affect melanoma invasiveness and adhesion (*Bell et al., 2014*). In our study, both miR-204 and miR-211 were down regulated in melanoma tissues, but miR-204 was positively associated

with tumor progression , miR-211 showed the opposite results. This result may be due to the small sample size of the GEO data set we selected. The reason for this situation needs further exploration. Early research of miR-510 did not examine its relationship with cancer, only its aberrant expression in irritable bowel syndrome (*Kapeller et al., 2008*). Researchers later discovered its relationship to cancer; for example, *Patnaik et al. (2010)* found that miR-510 can predict the survival of patients with local stage I non-small cell lung cancer after surgical resection. miR-510 also plays a role in promoting tumor growth and invasion in breast cancer (*Guo et al., 2013*). In our study, we found that the higher the expression of miR-510, the poorer the prognoses of melanoma patients. *Mairinger et al. (2014)* proposed that the expression of miR-513c in pulmonary neuroendocrine tumors was involved in tumor grade, and *Wang et al. (2015)* studied African-American patients with prostate cancer and acquired similar results. Previous studies also included miR-513c in the establishment of a nine-miRNA based uveal melanoma prognostic model (*Xin et al., 2019*). However, we missed including a related study on the mechanism of action of miR-510 and miR-513c in cutaneous melanoma.

To further verify the active mechanisms of these five miRNAs, we performed miRNA target gene predictions and found a total of 367 overlapping target genes. Interestingly, 222 of them were target genes derived from miR-25 (mainly enriched in the PI3K-Akt signaling pathway), 77 were target genes derived from miR-510 (mainly enriched in ubiquitin-mediated proteolysis), and the few remaining genes were target genes of other miRNAs. KEGG pathway analysis revealed that these target genes were mainly involved in the PI3K-Akt pathway, ubiquitin-mediated proteolysis, and cell adhesion. The most abundant genes, including PHLPP2, SGK3, CREB5, COL5A3, PTEN, ITGA5, ITGAV, COL27A1, CREB3L2, COL1A2, PPP2R5E, PIK3AP1, GNG2, and PIK3R3, were enriched in the PI3K-Akt pathway. The PI3K-Akt pathway (phosphoinositide 3-kinase-RAC-alpha serine/threonine-protein kinase) is involved in a variety of cellular processes in both normal cells and cancer cells, including cell survival, metabolism, transmigration, and proliferation (*Schadendorf et al., 2018*). For example, abnormal PI3K-Akt-mTOR signaling is one of the most common dysfunctions present in human cancers (*Janku, Yap & Meric-Bernstam, 2018*). The use of PI3K-Akt as a target in the treatment of cancer has been extensively researched (*Hamzehzadeh et al., 2018*). Phosphatase and tensin homolog (PTEN), a tumor suppressor gene, has been the focus of many studies. PI3K-Akt-PTEN signaling has been confirmed in previous studies of breast cancer and prostate cancer (*Jamaspishvili et al., 2018*; *Schadendorf et al., 2018*; *Yang, Polley & Lipkowitz, 2016*). In cutaneous melanoma, PTEN is downregulated, and this type of PTEN loss can increase T cell-mediated immunotherapy resistance (*Peng et al., 2016*). In our study, we showed that miR-25 may promote the development of cutaneous melanoma by downregulating the expression of PTEN, thus affecting the prognoses of melanoma patients. Feng et al. increased the sensitivity of hepatoma stem cells to TRAIL (Tumor necrosis factor (TNF) -related apoptosis-inducing ligand), reducing apoptosis by knocking out miR-25. This effect is also achieved through the PTEN/PI3K/Akt signal pathway (*Feng et al., 2016*). The adhesion-related genes obtained by KEGG analysis included ACTB, ITGA5, ITGAV, COL27A1, COL1A2, RAP1B, SHC1, COL5A3, PIK3R3, PPP1CC, and PTEN, and we found

that most of these genes were also involved in the PI3K signaling pathway. Degradation of the basement membrane and the extracellular matrix (ECM) is critical for the invasion and metastasis of malignant cells. ECM ligands not only control cell adhesion, migration, and actin cytoskeleton structure through signaling pathways, but they also control anchorage dependence, a key group of survival mechanisms (*Ciolczyk-Wierzbicka & Laidler, 2018*; *Multhaupt et al., 2016*). While COL27A1, COL1A2, and COL5A3 are encoded ECM components, experiments have found that the upregulation of COL1A2 in cancer can serve as a molecular basis for metastasis development (*Lin et al., 2016*). Integrins are the major cell-adhesive receptor; are key components of signaling molecules, mechanical transducers, and cellular migration mechanisms; and are involved in many aspects ranging from primary tumor to metastatic cancer progression (*Hamidi & Ivaska, 2018*) Integrin-alpha-5 (ITGA5) and integrin-alpha-V (ITGAV) are members of the integrin receptor gene family, closely related to the regulation of cancer growth and metastasis (*Chernaya et al., 2018*; *Morandi et al., 2016*). Ubiquitination is a widespread post-transcriptional modification, a selective marker that binds to protein aggregates and dysfunctional organelles, thereby promoting autophagy-dependent degradation. Autophagy can promote the growth of tumor cells by maintaining arginine-serine circulation (*Grumati & Dikic, 2018*; *Poillet-Perez et al., 2018*). *Yang et al. (2019)* showed that ubiquitin-mediated proteolysis by bioinformatics analysis may have potential as a prognostic and predictive marker for survival in patients with uveal melanoma.

In summary, the five screened miRNAs and their target genes were closely related to the occurrence and development of tumors, and the results proved reliable. Our established five-miRNA signature theoretically predicted the survival of melanoma patients, but further experiments are needed to determine the mode of action and mechanism of these miRNAs, and whether they can be used in a non-invasive diagnostic method.

## CONCLUSION

The five screened miRNAs and their target genes were closely related to the occurrence and development of tumors, and the results proved reliable. The identified five-miRNA signature may serve as a prognostic biomarker, or even as a potential therapeutic target, in cutaneous melanoma patients.

**Abbreviations**

| | |
|---|---|
| **miRNAs** | microRNAs |
| **DEMs** | the differential expression of miRNAs |
| **GEO** | Gene Expression Omnibus |
| **TCGA** | The Cancer Genome Atlas |
| **UM** | Uveal Melanoma |
| **DAVID** | The Database for Annotation, Visualization and Integrated Discovery |
| **GO** | Gene Ontology |
| **OS** | overall survival |
| **HR** | hazard ratio |
| **PI3K-Akt** | phosphoinositide 3-kinase-RAC-alpha serine/threonine-protein kinase |

| CM | cutaneous melanoma |
| ECM | extracellular matrix |

## ACKNOWLEDGEMENTS

We thank Chen Zhang, Yaling Li, and Xiaoqing Jian for their support over the past years.

### Funding

This study was supported by the Science & Technology Fund of Liaoning Province (grant no. 201501013). The funders had no role in study design, data collection and analysis, decision to publish, or preparation of the manuscript.

### Grant Disclosures

The following grant information was disclosed by the authors:
The Science & Technology Fund of liaoning Province: 201501013.

### Competing Interests

The authors declare there are no competing interests.

### Author Contributions

- Tao Lu conceived and designed the experiments, performed the experiments, analyzed the data, authored or reviewed drafts of the paper, approved the final draft.
- Shuang Chen analyzed the data, authored or reviewed drafts of the paper, approved the final draft.
- Le Qu conceived and designed the experiments, contributed reagents/materials/analysis tools, prepared figures and/or tables, approved the final draft.
- Yunlin Wang analyzed the data, prepared figures and/or tables, approved the final draft.
- Hong- duo Chen and Chundi He conceived and designed the experiments, authored or reviewed drafts of the paper, approved the final draft.

### Data Availability

The raw measurements are available in the Supplemental Files.

### Supplemental Information

Supplemental information for this article can be found online at http://dx.doi.org/10.7717/peerj.7831#supplemental-information.

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
