# Peer review of "Identification of a five-miRNA signature predicting survival in cutaneous melanoma cancer patients"

_PeerJ, doi:10.7717/peerj.7831_

## Round 0.1 · original submission · Major Revisions

The manuscript is aimed at demonstrating the usefulness of the evaluation of the expression levels of specific miRNAs as prognostic biomarkers for cutaneous melanoma. The manuscript is clear and well written.

Despite the idea is interesting, the investigation performed is not fully satisfactory. The authors must revise the manuscript according to the reviewers' suggestions.

Reviewer 1 ·

Basic reporting

The English language needs some revisions performed by an English native speaker. (e.g. Page 6 line 96, substitute the word “programme” with “program”, the same for Figure Legend 3; Page 6 line 86 miRNAs is not uppercase; Page 5 line 70 substitute the sentence “Melanoma is produced by the malignant transformation...” as follows “Melanoma is the result of the malignant transformation...”).
In the Introduction Section some information are lacking and some references are not appropriate for the contents of the manuscript (See General Comments).
Figures and tables are of good quality and clearly describe the results achieved by the authors. Raw data is correctly provided.

Experimental design

The manuscript proposed by Tao L and co-workers fits well with the aim of the journal. The scientific question proposed by the authors is clearly expressed and is aimed at demonstrating the usefulness of the evaluation of the expression levels of specific miRNAs as prognostic biomarkers for cutaneous melanoma. The scientific idea is very current and attractive because of the lack of effective prognostic biomarkers for cutaneous melanoma. Despite the idea is quite innovative and original, the investigation performed is not fully satisfactory because the authors analyze only one GSE Datasets excluding other datasets potentially interesting (see General Comments). In addition, some information about the differential and statistical analyses performed are lacking. Therefore, the description of methods could be improved.

Validity of the findings

The results obtained by the authors will pave the way to new experimental and observational study performed on in vitro, in vivo and clinical models in order to validate the prognostic significance of the five selected miRNAs. The data obtained would have been more significant if the authors had taken into account more GSE datasets of miRNAs expression. Overall, the presentation of the data is clear and of interest for the researchers involved in this field of study. The statistical analyses performed seem appropriate but it can be improved. Finally, the conclusion appears clear and summarizes well the main findings of the study.

Additional comments

Below are reported some minor/major revisions that the authors have to address:

1) In the introduction section, the authors should describe the importance of bioinformatics prediction analyses for the identification of putative diagnostic and prognostic biomarkers and describe other bioinformatics study performed on GEO DataSets and TCGA regarding the involvement of miRNA in cancer development. The authors should also explain why their study is different compared to those performed by other authors. For this purpose see the methods used in other similar studies:
- 10.18632/oncotarget.18222
- 10.1038/JID.2015.355
- 10.3892/ol.2015.4031
- 10.3390/cancers11050610

2) “Although the treatment of melanoma is constantly developing, the results are still unsatisfactory, and life-threatening if metastasis occurs, prompting clinicians to pursue new predictive markers and targeted therapeutic genes (Dyduch et al. 2017; Shain & Bastian 2016)” The references indicated for in the above sentences do not deal with this topic in detail. It is suggested to provide more appropriate references, for this purpose see:
- 10.3389/fphar.2018.01300
- 10.3390/healthcare2010001
- 10.3892/ijo.2018.4287

3) In the subheading “Acquisition of microarray data source and DEMs” of the “Materials & Methods” section, authors have to specify why they do not use the GEO2R tools provided by GEO DataSets for the differential analysis. Furthermore, the authors should specify why they selected the GSE35579 dataset. Have they applied inclusion or exclusion criteria for datasets selection? Clarify why the authors do not analyze other melanoma miRNA expression datasets, such as GSE62370 (tissues), GSE34460 (tissues), GSE18509 (tissues), GSE100508 (exosomes), GSE88727 (lymph nodes), GSE61741 (RNA from blood) and GSE20994 (RNA from blood). At least GSE62370, GSE34460 and GSE18509 should be included because built with expression data derived from melanoma tissue samples as the GSE35579 dataset.

4) How the GEO DataSets and TCGA miRNA expression data were compared and normalized?

5) In the first paragraph of the subheading “DEM identification and construction of a five-miRNA signature”, the authors talk about 186 differentially expressed genes of which 80 up-regulated and 105 down-regulated. Did the authors mean 186 differentially expressed miRNAs? Please check this paragraph and correct the errors, if any.

6) It is not clear how the authors select the five miRNAs. Are these miRNAs the most up-regulated or down-regulated? Did the authors only refer to the survival data for the choice of miRNAs?

7) Lines 243-246 page 10, sentences repeated.

Reviewer 2 ·

Basic reporting

The authors proposed to identify signature miRNAs to improve prognostic markers that can be useful therapeutic targets during the future therapy development. Their subject was clear without unambiguousity, and overall studies were performed well to answer their scientific questions. English language used throughout their manuscript was proper level to be published. Introduction provided enough background to understand what the authors aimed to in this project. Literatures were well referenced by using most relevant and up-to-date works. Figures were well-organized with relevance to their findings, and the figure legends were well written.

Experimental design

The authors used GEO database to identify DEMs from melanoma patients, and crossed them to TCGA database to correlate their contribution to prognosis. Enough statistical consideration was made to rule out any unreasonable observations.

Validity of the findings

The authors identified five DEMs that were most up or down-regulated in the database, and showed the clinical relationship by comparing the expression levels to survival rates in melanoma patients. Based on their findings, it seems those five DEMs can serve as prognostic markers in clinical application. Although they did not perform wet laboratory studies to validate the functional role of the five DEMs, the authors justified well their findings by critically discuss their findings to a great volume of references in the discussion. Therefore, their findings can be described as a groundwork for future studies where the five DEMs should be further studied or considered as signature miRNAs in melanoma development and progression.

Additional comments

The authors has redefined the signature miRNAs in melanoma patients by re-analyzing two databases GEO and TCGA with enough statistical consideration. Overall research was performed well with clear description about their methodologies and findings with enough figures. Although they did not perform wet laboratory studies to validate the functional role of the five DEMs, the authors justified well their findings by critically discuss their findings to a great volume of references in the discussion. Therefore, their findings can be described as a groundwork for future studies where the five DEMs should be further studied or considered as signature miRNAs in melanoma development and progression. The manuscript will need only minor work on the formatting. For examples, many places of period or comma has space gaps after the last letter in the sentence.

---

## Round 0.2 · accepted · Accept

The manuscript has been improved and is acceptable for the publication in PeerJ

Reviewer 1 ·

Basic reporting

The revised version of the manuscript meet the standards of the journal, also the English language was improved.

Experimental design

The authors provided more details about the experimental design and workflow. They well addressed all the reviewers' comments.

Validity of the findings

The achieved results are satisfactory and of interest for the readers.

Additional comments

The authors well addressed all the reviewers' comment.